# Intervention of Physical Activity for University Students with Anxiety and Depression during the COVID-19 Pandemic Prevention and Control Period: A Systematic Review and Meta-Analysis

**DOI:** 10.3390/ijerph192215338

**Published:** 2022-11-20

**Authors:** Qingyuan Luo, Peng Zhang, Yijia Liu, Xiujie Ma, George Jennings

**Affiliations:** 1School of Wushu, Chengdu Sport University, Chengdu 610041, China; 2School of Foreign Languages, Xi’an Jiaotong University, Xi’an 710100, China; 3Chinese Guoshu Academy, Chengdu Sport University, Chengdu 610041, China; 4Cardiff School of Sport and Health Sciences, Cardiff Metropolitan University, Cardiff CF23 6XD, UK

**Keywords:** COVID-19 epidemic, physical activity, university students, anxiety, depression, meta-analysis, systematic review

## Abstract

(1) Background: Although physical activity has been widely recognized as an effective way to improve anxiety and depression, we lack a systematic summary of research on improving anxiety and depression during the COVID-19 pandemic. The study aims to systematically analyze how physical activity impacts on this situation in college students during COVID-19. (2) Methods: Both Chinese and English databases (PubMed the Cochrane Library, EMBASE, Web of Science, Scopus, Chinese National Knowledge Infrastructure, Wanfang) were analyzed. All the randomized controlled trials (RCTs) about physical activity intervention for this were included. We received eight eligible RCT experiments before the retrieval time (4 October 2022) in the meta-analysis. (3) Results: Physical activity benefits for college students with significant anxiety were (SMD = −0.50; 95% CI = −0.83 to −0.17; I^2^ = 84%; *p* < 0.001; Z = 2.98;) and depression (SMD = −0.62; 95% CI = −0.99 to −0.25; I^2^ = 80.7%; *p* < 0.001; Z = 3.27). Subgroup analyses showed physical activity of different intensities significantly impacted on improving college students’ depression and anxiety, but physical activity of 6 < 9 Mets intensity had a greater effect on anxiety than on depression. Interventions of eight weeks or less performed better than those of over eight weeks while interventions less than four times per week had a significant effect on improving the situation. The overall effect of a single intervention of 30 min was more effective than one of over 60 min. (4) Conclusion: Physical activities can effectively improve the situation of anxiety and depression for college students during the COVID-19 pandemic. However, a higher quality RCT experiment is needed to prove it.

## 1. Introduction

COVID-19 is a contagious respiratory disease caused by a new type of coronavirus called Severe Acute Respiratory Syndrome-Coronavirus-2 (SARS-CoV-2) [1]. By 11 October 2022, 620 million confirmed cases and more than 6.54 million deaths had been reported globally [2]. The spread of COVID-19 has had serious negative impacts on the global economy (such as aviation, tourism, construction) [3,4,5,6,7], and society (e.g., domestic violence, child abuse) [8,9,10,11]. The most notable aspects have been populations’ mental health (such as anxiety, depression, insomnia, anger) [12,13,14]. Although government efforts have largely relied on physical space interventions such as social separation and self-isolation to control COVID-19, aiding the prevention of the spread of COVID-19, they have undoubtedly resulted in a reduction in face-to-face interactions [15]. Consequently, COVID-19 has drastically decreased the population’s time and spatial range of physical exercise, and regional athletic events have been canceled or postponed [16].

Early in the COVID-19 pandemic, researchers reported that restricted range of motion and physical activity in quarantined residential areas could lead to the incidence of additional psychological problems such as depression, anxiety, suicidal tendency, and intentional self-harm [17,18,19]. In addition, the fear of COVID-19 during quarantine periods can also heighten negative emotional responses in individuals, which may potentially be translated into a range of mental health and behavioral disorders [20,21,22,23,24]. As for university students, they are facing the double pressure of both employment and academic stress at the same time [25,26]. During the long-term prevention and control of the COVID-19 pandemic, university students’ lifestyles and behavioral characteristics have undergone tremendous changes, due to such factors as the reduction of time for sports activities, the limitation of sports activities, an increase in sedentary time and the use of electronic screens, and the deterioration of mental health [27,28]. A survey of university students from Bangladesh during the COVID-19 pandemic indicated that 40% of participants had moderate to severe anxiety, 72% had depressive symptoms, and 53% had a moderate to poor mental health status. A survey from Greece showed that there was a 2.5–3 times increase in clinical cases of depression and a nearly 8 times increase in suicidal tendencies in Greek university students [21,29]. Research reports from the United Kingdom, China, and the United States have also pointed out the serious impacts of the COVID-19 pandemic on the mental health of university students in their countries (such as depression, anxiety, suicidal tendency) and other health problems [30,31,32,33]. At the same time, long-term online courses during the COVID-19 pandemic outbreak and prevention have led to the reduction of physical activity time and the increase in sedentary time for university students, which in turn have caused problems such as depression, anxiety, weight gain, boredom, learning challenges, and insecurity of performance. These problems have further aggravated university students’ mental health issues [34,35,36,37,38]. In terms of emotion regulation, some mental illnesses are considered to be an impairment. Emotion regulation, as a key to mental health, can effectively reduce the incidence of stress-related psychiatric disorders [39,40]. According to a study, during COVID-19, lower levels of negative emotion regulation were positively associated with overall negative mood. That is to say, individuals with lower negative emotion regulation experienced a higher reactivity of negative emotion, and this association was closely related to the level of participation in exercise [41]. There is also a strong correlation between physical activity and university students’ academic pressure and internal motivation. A certain degree of physical activity can significantly reduce students’ academic pressure and improve students’ internal motivation for learning. Moreover, there was a significant association between physical activity and emotion regulation. Comparing two groups of college students, those who met the minimum level of physical activity showed higher levels of emotion regulation and a lower incidence of mental health illness than those who did not [42]. In addition, some studies have pointed out that during COVID-19 isolation, cognitive behavioral therapy (CBT) as a non-physical activity can also effectively relieve students’ emotional disorders (fear, stress, depression, etc.) [43]. The World Health Organization recommends that adults who sit at home for a long time should do at least 150 min of moderate intensity physical activity or at least 75 min of intense intensity physical activity within a week to reduce the risk of depression and enhance mental health [44]. However, it is worth noting that during COVID-19, due to social isolation, online physical activity and exercise prescription became increasingly popular. Many people have changed their previous physical activity methods and engaged in online-guided indoor exercise, which seems to be the safest exercise method during COVID-19 [45].

Several pieces of research have shown that physical exercise can improve the physical and mental health of individuals of different ages [46,47,48,49,50]. Some studies have pointed out that physical activity can regulate the circadian rhythm of individuals, activate their metabolism and physiological activity, improve their emotional disorders, increase levels of serotonin and noradrenaline, and improve their physical and mental health [51], especially for those with depression, anxiety, and other mental diseases [52,53,54,55]. A follow-up experiment reports that adolescents with more positive attitudes toward exercise and fitness may be more physically and mentally active in 5 and 10 years [56]. Regular exercise is also more conducive to maintaining public mental health [57]. Several pieces of research conducted during the COVID-19 pandemic revealed that physical activity benefits university students’ mental health [58,59,60,61,62,63,64,65]. However, it is undeniable that compared to the situation before the outbreak of COVID-19, university students’ physical activity time has decreased significantly by about 48% to 61% [66,67]. Some studies have pointed out that during the COVID-19 prevention and control period, the intensity of physical activity and negative emotions have had a U-shaped relationship [68]. However, a study from the United Kingdom showed that the deterioration of the mental health of university students during the COVID-19 prevention and control period was not related to increased stress and changes in physical activity [69]. In addition, some studies have pointed out that there is no significant correlation between mental health (anxiety, depression, and stress) and the physical activity of university students during the COVID-19 prevention and control period [70,71].

It can be seen that the specific impacts of physical activity on the mental health (depression, anxiety) of university students during the COVID-19 prevention and control period have not been clearly demonstrated. Therefore, this study seeks to answer a research question: How did physical activity impact the mental health (depression, anxiety) of university students during the COVID-19 prevention and control period? In order to answer this question, we conducted a meta-analysis to evaluate the effects of physical activity. Consequently, it is organized as follows. First, we address the materials and methods used for the meta-analysis. Second, we present the results of the analyses. Third, we discuss the results and link them to previous studies. Fourth, we address the limitations of the study and then conclude.

## 2. Materials and Methods

### 2.1. Search Strategy

This systematic review followed the guidelines of the Preferred Reporting Items for Systematic Reviews and Meta-Analyses (PRISMA statement). The literature search was conducted in Chinese and English databases including PubMed the Cochrane Library, EMBASE, Web of Science, Scopus, Chinese National Knowledge Infrastructure, and Wanfang Data, with PICOS (participants interventions comparisons outcomes) as a retrieval reference [72]. Databases were searched using the following terms: (sports movement or physical exercise or sports activities or sport or motor or athletic sports or exercise or physical activity or taiji or Tai chi or baduanjin or yoga) AND (psychology or psychological or mental health or depression or anxiety or mental illness or mental disorder or stress or emotional health) AND (Corona-virus or novel corona-virus or COVID-19 or COVID-2019 or 2019-Ncov) AND (university student or college student or undergraduate or academician or higher education students). Only studies published between 1 January 2020 and 4 October 2022 were included.

### 2.2. Inclusion and Exclusion Criteria

The inclusion and exclusion criteria of the literature are detailed in Table 1. If there was a disagreement between two authors during the literature selection process, the third author (X.M.) would be a participant in problem solving.

### 2.3. Study Selection and Data Extraction

Two reviewers (L.Q. and Z.P.) independently performed the literature review, data extraction, and crosschecking. If the results of the chosen article were elaborated in a graphical format, efforts were made to request their authors for numerical data. Extracted data on the following variables were: 1. basic information of the studies (such as first author, publication time, country, sample size); 2. basic characteristics of study participants (such as age, gender, education level); 3. intervention measures (such as intervention items, duration, number of times per week, time of each intervention); 4. tools for assessing mental health; and 5. outcome measures (such as depression, anxiety, and other psychological problems). Inconsistencies were resolved by a third independent reviewer and group discussion, as required (X.M.).

### 2.4. Quality Assessment of Eligible Study

Two authors (Q.L. and P.Z.) utilized a tool offered by the Cochrane Collaboration (Cochrane RoB1.0) [74]. We assessed seven key domains: allocation concealment, sequence generation, blinding of participants and personnel, incomplete outcome data, blinding of outcome assessment, selective outcome reporting, and other sources of bias. For each included study, the risk of bias was classified as low, unclear, or high. Discrepancies in assessments were resolved by a third party (X.M.).

### 2.5. Study Analysis Method

Stata 17.0 was used for heterogeneity test, data merging, forest plot, and risk of bias assessment. The outcomes in the literature review were continuous outcome variables, which were inconsistent with the units of the outcome indicators in the original design. For continuous outcomes, we calculated the standardized mean difference (SMD) and 95% confidence intervals (95% CI). -SMD- < 0.1 was a small effect size, 0.10 ≤ -SMD- ≤ 0.34 was a small effect size, 0.35 ≤ -SMD- ≤ 0.64 was a medium effect size, 0.65 ≤ -SMD- ≤ 1.19 was a large effect size, and -SMD- ≥ 1.20 was a very large effect size [75]. The heterogeneity between studies was judged, in which *p* < 0.05 was taken as the significant level and I^2^ statistic was used to assess heterogeneity. I^2^ (0%), I^2^ (≥25%), I^2^ (≥50%), and I^2^ (≥75%), respectively, indicated no heterogeneity, mild heterogeneity, moderate heterogeneity, and high heterogeneity. When I^2^ ≥ 50%, the random-effects model was used to combine the data. When I^2^ < 50%, the fixed-effects model was used. If a crossover design was found in the articles, the results in the primary stage before the crossover design were selected. If there was more than one physical activity group studied in the articles, the groups were combined to avoid the loss of data and potential analysis [76].

## 3. Results

### 3.1. Studies Selection

In the database, we retrieved 2811 publications in total. In the preliminary screening, we eliminated 2709 studies by removing duplicate publications and reading titles and abstracts, including duplicate publications (*n* = 127), and unrelated studies (*n* = 2480). The remaining 102 publications were further screened through full text reading and only 8 were singled out for meta-analysis, as the other 94 studies or experiments were not related to the design of the study due to such factors as lack of physical activity for intervention (*n* = 43), the time of the experiment before the outbreak of COVID-19 (*n* = 4), and not targeting university students as the study objects (*n* = 47). Detailed analysis and characteristics about the meta-analysis of the eight studies are shown, respectively, in Figure 1 and Table 2.

### 3.2. Characteristics of Eligible Studies

The eight studies included in the research were distinctive in their publication. Two of them were published in Chinese [58,59], and six in English [60,61,62,63,64,65]. They covered research held in China [58,59], the United States [60], Saudi Arabia [62], India [63], Turkey [64], and Belgium [65]. The sample size ranged from 28 to 387, with a total of 1195 participants, including 607 participants in the physical exercise groups and 588 participants in the control groups. The ages ranged from 18 to 30 [58,60,61,64,65]. Among them, two studies did not report the age range of the university students for their research [59,60], four studies showed that women accounted for the majority of the research objects [62,63,64,65], two studies showed that the proportion of men and women was almost equal [58,61], one study only included female university students [59], and one did not report the gender of the study subjects [60], and five studies reported the inclusion criteria for mental health problems (depression, anxiety) [58,59,63,64,65]. All the included research types were RCT experiments and the mental health scale [62,64,65], three unreported Cronbach’s α except for the coefficient, and all the others were in the range of 0.79~0.93 (if the reliability coefficient of the scale is greater than or equal to 0.70, the reliability of the scale is considered good) [77].

### 3.3. Interventions and Controls

Table 3 summarizes the characteristics of the physical activity intervention measures in all the included trials. There are four experiments based on aerobic physical activity [59,62,64,65], two experiments based on Baduanjin, a kind of Chinese traditional sport [58,61], one experiment based on Yoga [60], and one experiment based on body weight training + WBV vibrator [63]. However, there are some differences in all the types of physical activity. Therefore, the reported activity type or intensity is converted into a metabolic equivalent (MET) to estimate the activity intensity of the same standard [78,79]. Among them, the type of physical activity of four experiments is MODERATE (3 < 6 Mets) [58,60,61,64], and the type of physical activity of four experiments is VIGOROUS (6 < 9 Mets) [59,62,63,65]. The duration of the intervention ranges from 4 to 18 weeks, more than two to five times a week, and the duration of a single exercise session ranges from 10 to 90 min. There are five physical activity experiments with supervision [58,59,62,64,65], five experiments with physical activity coaches [58,59,61,64,65], two experiments with group physical activity intervention [58,59], and six experiments with individual physical activity intervention [60,61,62,63,64,65]. Among them, the control group of four experiments is the non-active normal life group [58,59,64,65], one is the waiting control group [60], and the other three are the control groups of learning health knowledge and CBT intervention, and the control group of not conducting body weight training on the vibrator [61,62,63].

### 3.4. Risk of Bias

Figure 2A,B shows the bias risk assessment of the included studies. All included experiments had the bias risk of “unclear” [58,59,60,61,62,63,64,65]. Four studies reported the generation and allocation hiding of random sequences [60,62,63,65]. In the allocation hiding, two studies used computer-generated random number sequences of REDCap [60,63], one assigned a unique computer-generated random code to each participant by simple randomization [63], and one assigned a secret allocation replacement interval [65]. One study [59] divided depression and anxiety into different groups at the time of grouping, so it presented high risk in random sequence and allocation concealment.

### 3.5. Meta-Analysis of Outcome Indicators

#### 3.5.1. Physical Activity Compared to No Intervention Control

Meta analysis included a total of eight experiments (*n* = 1196) [58,59,60,61,62,63,64,65], including multiple experimental groups [59]. The effects of different experimental groups were combined when included in the forest map analysis. The research experiment used a cross design [60]. When included in the forest map analysis, the pre-cross stage between the control conditions and the experimental group was selected. The evaluation with the physical activity experimental group and the control group without intervention was conducted using the standardized mean difference (SMD). Because the heterogeneity was higher, I^2^ > 50%, the random model was used for analysis. In eight experiments, the depression and anxiety state after physical activity intervention were measured. Compared with the control group without physical activity intervention, physical activity brought about a statistically significant improvement of medium effect [58,59,60,61,62,63,64,65]. For the improvement of anxiety symptoms (SMD = −0.50; 95% CI = −0.83 to −0.17; I^2^ = 84%; *p* < 0.001; Z = 2.98, Figure 3); for the improvement of depressive symptoms (SMD = −0.62; 95% CI = −0.99 to −0.25; I^2^ = 80.7%; *p* < 0.001; Z = 3.27, Figure 4). The heterogeneity of the included experiments is high, which may be due to the diversity of the control interventions and measurement tools used.

#### 3.5.2. Regression Analysis

Covariates (such as Met intensity, frequency of physical activity per week, duration of physical activity per day, and duration of intervention weeks) can be influencing factors for relieving depression and anxiety through physical activity. It has been pointed out that in the same period, university students in different countries had almost the same Met values [80]. Met can be used to measure the load and stimulation of different physical activities on the human body [79]. Therefore, Met is included in the regression analysis of subgroups as a strength indicator. Table 4 and Table 5 list the covariate regression analysis results of physical activity for relieving depression and anxiety. Accordingly, physical activity has no significant effect on anxiety. Met intensity (95% CL—1.727567 to 2,211,914, *p* = 0.091), weekly physical activity frequency (95% CL—4,837,114 to 3,423,562, *p* = 0.0624), and daily physical activity time (95% CL—559,598 to 2,805,465, *p* = 0.468), and duration (95% CL—6,970,513 to 1.18863, *p* = 0.468). For depression, Met intensity (95% CL-2.670032 to 1.055778, *p* = 0.203), for weekly physical activity frequency (95% CL—1.612992 to 3.568875, *p* = 0.246), for daily physical activity time (95% CL—1.274012 to 1.276941, *p* = 0.997), and for duration weeks (95% CL—2.05842 to 2.121378, *p* = 0.954).

#### 3.5.3. Sub-Group Analysis

In order to further discuss the impact of physical activity intervention on university students’ anxiety and depression during the epidemic prevention and control period, we divided the subjects into four subgroups according to the Met intensity, the number of physical activity sessions per week, the duration of physical activity, and the number of weeks of physical activity in all the included literature, considering the differences in the research characteristics of each experiment (see Table 6 and Table 7 for details). The results of the subgroup analysis are as follows regarding anxiety disorder and depression: (1) Physical activity of different intensity can help improve anxiety and depression symptoms, and physical activity of 6 < 9 Mets intensity can improve anxiety and depression more than physical activity of 3 < 6 Mets. (2) In terms of the duration of physical activity per week, the effect of interventions with physical activity less than four times per week on anxiety and depression is greater than that for more than four times per week. (3) For the duration of physical activity per day, physical activity for or less than 30 min for university students with anxiety disorders can bring more significant effects than that for 30-60 min, or that for over 60 min, and the single intervention times of 30 min and less and more than 60 min have significant effects for depression. (4) For the duration of weeks, interventions for less than eight weeks and more than eight weeks can bring about significant effects in improving anxiety and depression, but the intervention effects of less than eight weeks are better.

## 4. Discussion

The systematic review and meta-analysis of this study aim to sort and analyze the related experiments on physical activity for university students during the prevention and control of COVID-19, and to understand the mechanism and effects of physical activity on university students’ mental illness during the prevention and control of COVID-19, so as to provide an exercise prescription for relieving university students’ mental health during COVID-19. The results show that physical activity is beneficial to alleviate the mental health disorders of university students during the COVID-19 prevention and control period, and physical activity of different intensities can improve the mental illness of university students to varying degrees (Anxiety; SMD = −0.50; 95% CI = −0.83 to −0.17, *p* < 0.05; Depression; SMD = −0.62; 95% CI = −0.99 to −0.25; *p* < 0.001).

The results of this systematic evaluation are consistent with the conclusions of the previously published systematic evaluation, which evaluates the beneficial effects of physical activity on depression and anxiety during the COVID-19 pandemic [81,82,83,84]. At the same time, the research of Liuyang et al. may have significant heterogeneity [59]. Compared with other experiments included in the study, this study divided depression and anxiety into different groups at the time of experimental grouping. Through the analysis of the study, it is found that compared with other single studies, this study has a research design of multiple experimental groups, and different experimental groups have different project strengths. From the perspective of experimental objects, we found that the participants in the experiment were all female students.

Subgroup analysis shows that physical activity of a certain intensity can improve university students’ anxiety and depression, but it should be noted that Vigorous (6 < 9 Mets) physical activity is superior to Moderate (3 < 6 Mets) physical activity in improving university students’ anxiety and depression, and this finding is consistent with the research results during COVID-19 [85], but contrary to the research results during non-COVID-19 [86]. This may be because the isolation measures of COVID-19 have changed university students’ living habits and restricted their physical activity patterns (such as the increase in sedentary time, and the reduction of the scope of spatial activities) [87]. Our subgroup analysis results also show that the improvement effect of less than four interventions in a week on anxiety and depression shows a significant state, while more than four interventions do not show a significant state, which is also consistent with the existing research results [88], which may be because physical activity and mental health show a U-shaped trend; that is, too much physical activity will weaken the resilience of isolated people during COVID-19 (i.e., the ability to recover and maintain adaptive behavior after stress events) [89]. The subgroup analysis of the duration of one intervention showed that ≤30 min showed significant improvement in anxiety and depression, and ≥60 min physical activity intervention time only improved depression, not anxiety. Our research results confirm the previous prospective cohort report, which pointed out that three times a week with moderate or higher intensity (at least 15 min each time) can significantly relieve mental health [90]. The duration of single exercise ≥60 min was significantly related to the lower risk of depression [91]. In addition, subgroup analysis showed that in the number of intervention weeks for the improvement of anxiety and depression symptoms, the effect of an intervention time of less than eight weeks would be greater than that of an intervention for eight weeks, although both I^2^ were greater than 50%, and with the increase in intervention time, the relief effect of anxiety would be significantly reduced. A meta-analysis on Baduanjin’s role in relieving mental illness also pointed out a similar result; that is, there is a negative correlation between practice time and the decline of anxiety level [92]. This may be because long-term strenuous exercise may make people more likely to suffer from upper respiratory diseases and affect cellular immune function [93], thus offsetting the improvement effect of physical activity on depressive symptoms.

In the study of physical activity during COVID-19 on mental health, it was pointed out that physical activity can improve negative emotions such as long-term home isolation, fear of being infected by viruses, and students’ anxiety about learning caused by COVID-19 [94,95]. The interaction between emotion regulation and emotional reactivity can also be modulated by a certain amount of physical activity. Moreover, physical activity can effectively strengthen individual brain regions and large-scale neural circuits to improve emotional and behavioral regulation, enhance the self-regulation of emotions, and motivate individuals to adopt more adaptive emotion regulation strategies to cope with negative emotions during COVID-19 [96,97]. Some studies have pointed out that a certain degree of physical activity can significantly improve the immune function and reduce the incidence rate, increase the blood concentration of soluble angiotensin-converting enzyme 2 (ACE2), which has a protective effect against SARS-CoV-2 infection [98], and improve the sleep disorder caused by neuron damage caused by SARS-CoV-2 infection [99]. In addition, exercise leads to a significant increase in sympathetic nervous system activity and catecholamine secretion. It may potentially regulate the secretion of melatonin. As a body hormone, melatonin can significantly improve cardiovascular function, enhance the adaptability of skeletal muscle, and protect the health of the body [100]. In an experiment to control sleep time, it is shown that a certain amount of melatonin hormone supplement can improve the negative effects of physiological functions such as the prolonged reaction time caused by insufficient sleep, the decline of anaerobic exercise function, the increase in blood lactic acid level, and alleviate psychological and emotional disorders such as depression and anxiety [101]. This study also confirmed the improvement effect of physical activity on university students’ specific dimensions of mental health (anxiety, depression) during the prevention and control period of COVID-19. However, the impact on the internal mechanism of the human body has not been involved in this study. Therefore, the specific impact of physical activity under different intensities on the internal mechanism of the human body should be discussed in future research to provide the best exercise prescription to alleviate mental health during COVID-19.

In general, this research summarizes the different degrees of improvement of physical activity for university students’ mental health during the control and prevention of COVID-19, and also confirms the important role of physical activity for university students during the period. However, we need more randomized controlled experiments to strengthen this hypothesis.

## 5. Limitations

Although the studies included in this systematic meta-analysis were conducted in different countries such as China, the US, Belgium, Turkey, and Saudi Arabia, one third of the experiments were conducted in China. According to some research, the prevalence of anxiety and depression are in line with the pandemic, but it remains stable in countries with great epidemic prevention and control. Countries with good epidemic prevention and control have low risks of infection, as well as incidents of anxiety and depression, and so do university students [102,103]. Therefore, there may be some differences in the results, as well as ethnic differences and cultural differences, and other factors may affect the research results and limit the universality of the research results. In addition, many factors may also affect the results: for example, the overall quality of the evidence base for the main analysis is not high, the literature that can be included in the systematic analysis is limited, and the experimental intervention is online intervention. Moreover, in the subgroup analysis, due to the insufficient number of trials included in the analysis, our subgroup research results were limited by the lack of observation and motivation, which led us to be unable to investigate some important factors, including the lack of follow-up of the impact of physical exercise on the improvement of university students’ psychological diseases during COVID-19 and the results of the improvement of university students’ physical health after physical activity intervention. In addition, it should be noted that this experiment is an online intervention experiment. Although the experimenter carries monitoring equipment, the absolute accuracy of Met packet conversion is still lacking. Therefore, in future research, a variety of intervention designs and means should be used for comparison, such as the online and offline results of the same intervention means or the intervention of different physical activity intensities of the same intervention population.

## 6. Conclusions

The meta-analysis in this research indicates that physical activity has a significant impact on mental health improvement among university students with anxiety and depression during the control and prevention of COVID-19. Specifically, physical activity with an intensity of 3 < 6 Mets and 6 < 9 Mets can significantly improve university students’ anxiety and depression, but physical activity with an intensity of 6 < 9 Mets can bring about better improvement than physical activity with an intensity of 3 < 6 Mets. The effect of intervention for less than eight weeks will be greater than that of intervention for more than eight weeks, and with the increase in the number of intervention weeks, the relief effect on anxiety will be significantly reduced. The improvement effect of less than four interventions in a week on anxiety and depression showed a significant state, while more than four times did not show a significant state. Physical activity intervention of ≤30 min showed significant improvement in anxiety and depression, but ≥60 min physical activity intervention time only improved depression, not anxiety. It is necessary to note that emotion regulation and emotional reactivity are two essential elements for mental health, and that higher levels of emotion regulation and emotional reactivity, which are closely related to physical activity, can bring better mental health to individuals. The findings of this study theoretically enrich the dimension of emotion regulation theory, namely the mediating mechanism of physical activity between emotion regulation levels and emotion reactivity, and further support the idea that physical activity can improve emotion regulation levels, which has realistic significance for improving the mental health of college students during COVID-19.

It should be noted that the specific impact of the internal mechanism of the human body and the follow-up results after the intervention have not been obtained in this study. In addition, the quality and quantity of the methodology included in the test limit our ability to make conclusions about its effectiveness. In the future, larger range and higher quality RCT experiments are needed to verify the current research results.

## Figures and Tables

**Figure 1 ijerph-19-15338-f001:**
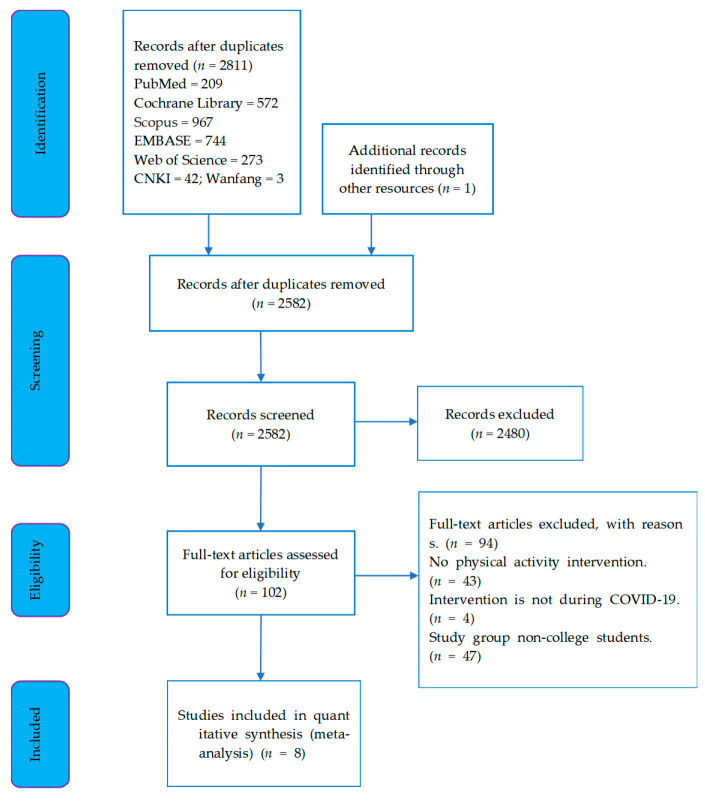
Flow of study selection.

**Figure 2 ijerph-19-15338-f002:**
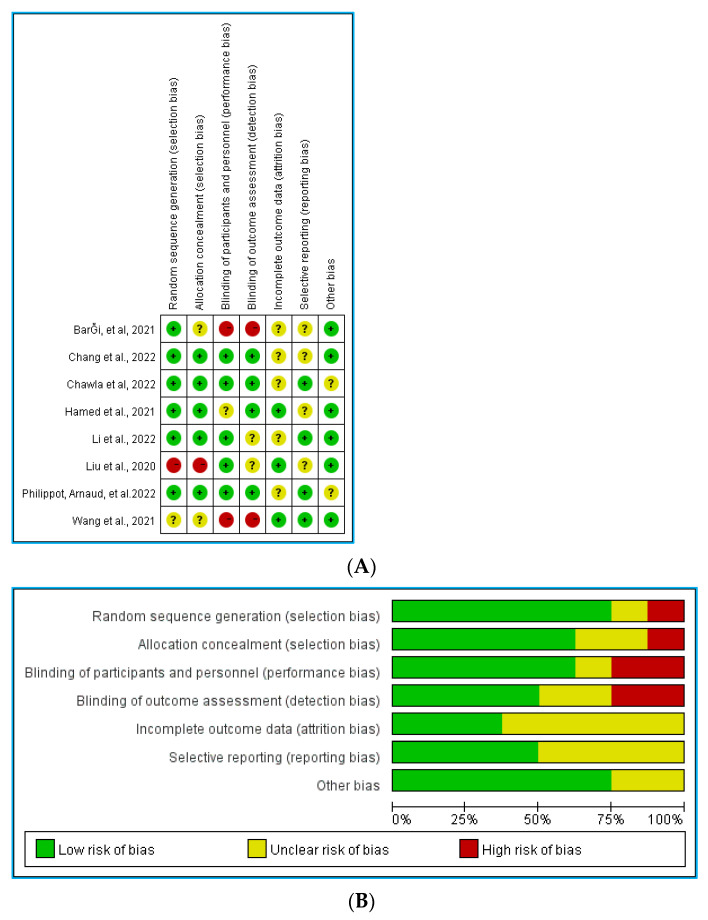
(**A**) Risk of bias ratings. (**B**) Risk of bias graph: percentage of trials receiving low, unclear, or high risk of bias rating for each domain [58,59,60,61,62,63,64,65].

**Figure 3 ijerph-19-15338-f003:**
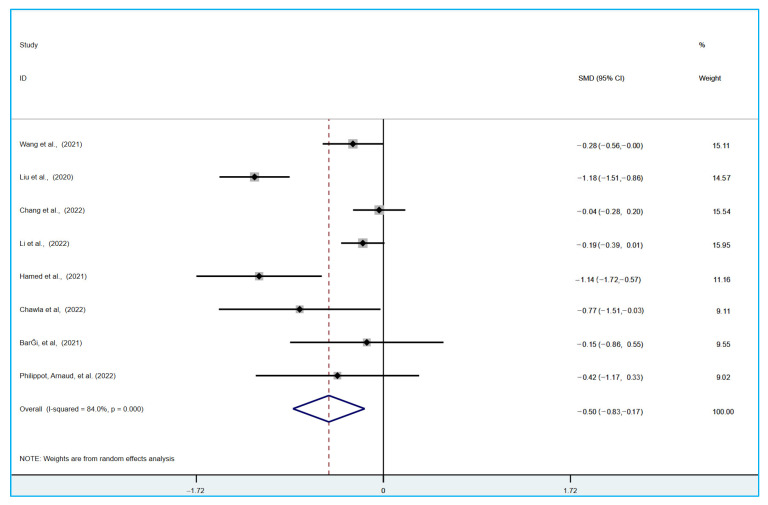
The Influence of Physical Activity on University Students’ Anxiety Disorder [58,59,60,61,62,63,64,65].

**Figure 4 ijerph-19-15338-f004:**
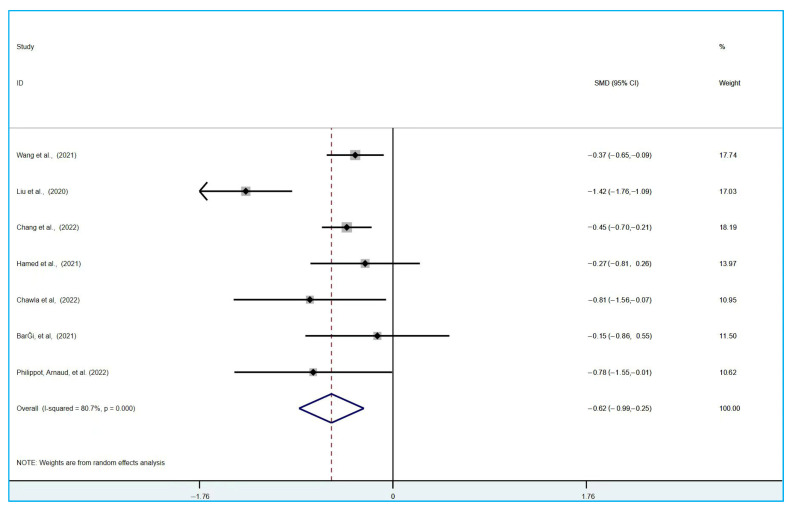
The Influence of Physical Activity on University Students’ Depression [58,59,60,61,62,63,64,65].

**Table 1 ijerph-19-15338-t001:** Inclusion and Exclusion.

Inclusion Criteria	Exclusion Criteria
1. Studies with Controlled Experimental Groups	1. Irretrievable literature
2. Studies include measures of depression and anxiety	2. Repeated research
3. The research object is university students	3. Review and observational or cross-sectional studies
4. All physical activity interventions are eligible, i.e., “any physical movement produced by skeletal muscle that results in energy expenditure above resting levels” [73]	4. Report with no valid results
5. Articles published between 1 January 2020 and 4 October 2022	
6. Experiments conducted during the outbreak of COVID-19	

**Table 2 ijerph-19-15338-t002:** Included trial characteristics.

Trial ID	Country	N	Age Mean (Range)	Gender, % Female	T	C	Recruitment Setting	Depression and Anxiety Inclusion Criteria	Scale Validity Value	Depression and Anxiety Outcome Measures	Research Type
Wang et al., 2021 [58]	China	200	19–21	51	100	100	College freshman	Depression ≥ 3	SCL Cronbach’s coefficient α = 0.79~0.99	SCL-90	RCT
Liu et al., 2020 [59]	China	192	NR	100	128	64	Undergraduate	Anxiety Score ≥ 50 Depression Score ≥ 50	SAS Cronbach’s coefficient α = 0.912 SDS Cronbach’s coefficient α = 0.818	SAS SDS	RCT
Chang et al., 2022 [60]	America	273	NR	NR	114	159	Undergraduate students aged ≥18 attending a 4-year college in the U.S. who did not graduate before May 2020	NR	PHQ-4 Anxiety Cronbach’s coefficient α = 0.83, Depression = 0.82 PANAS positive Cronbach’s coefficient α = 0.90, negative = 0.87 PSS-10 Cronbach’s coefficient α = 0.89	PANAS PHQ-4 PSS-10	RCT
Li et al., 2022 [61]	China	387	20–30	49.8	195	192	1. Between 20 and 30 years old. 2. Study or work in school for more than two months	NR	CAS Cronbach’s coefficient α = 0.793. PWBS Cronbach’s coefficient α = 0.93	CAS PWBS	RCT
Hamed et al., 2021 [62]	Saudi Arabia	54	18–25	70.3	27	27	Undergraduate	NR	NR	DASS21	RCT
Chawla et al., 2022 [63]	India	30	23–26	70	15	15	Undergraduate	Depression > 10 Anxiety > 8	DASS Anxiety Cronbach’s coefficient α = 0.84 Depression = 0.91	DASS SF-36	RCT
BarĞi, et al., 2021 [64]	Turkey	31	18–21	90.3	15	16	1. Undergraduate 2. aged ≥18 years 3. Having a smartphone or pedometer	Depression > 14 Anxiety > 8	NR	PA BAI BDI QOL, SF-36	RCT
Philippot, Arnaud, et al., 2022 [65]	Belgium	28	19–25	89.2	13	15	1. Undergraduate 2. Between 18 and 25 years old.	GAD-7 ≥ 5	NR	DASS-21 GAD-7	RCT

SCL-90 = Symptom Checklist-90; SAS = self-rating anxiety scale; SDS = self-rating depression scale; PSS-10 = 10-item Perceived Stress Scale; PHQ-4 = Anxiety and Depression brief questionnaire; PANAS = Positive and Negative Short-Form 10-item scale; CAS = Coronavirus Anxiety Scale; PWBS = Mental Health Scale, including six subscales; DASS21 and DASS = Depression, Anxiety and Stress scale; SF-36 = Depression Anxiety Stress Scale; PA = International Physical Activity Questionnaire-short form; BAI = Beck Anxiety Inventory; BDI = Beck Depression Inventory; QOL = The World Health Organization Quality of Life, WHOQOL (Short Form-36); GAD-7 = Generalized anxiety disorder scale; NR = not-reported or unclear; T = test group; C = Control group.

**Table 3 ijerph-19-15338-t003:** Characteristics of physical activity interventions from included trials.

Trial ID	Physical Activity Arms and Content	Aerobic/ Resistance	Setting	Duration (Weeks)	Session (Min)	Sessions per Week	Intensity (MET) *	Control Arm
Wang et al., 2021 [58]	1. Baduanjin	NR	S Q G	18 weeks	90 min	3	Mod (3 < 6)	NT
Liu et al., 2020 [59]	1. physical training@ HR (60~69%) 2. Taijiquan	Aerobic	S Q G	5 weeks	40 min	3	Vig (6 < 9)	Prevented from doing sports
Chang et al., 2022 [60]	1. Yoga Namaskar 2. Nadi Shuddhi	NR	NR NR I	12 weeks	40–45 min	3	Mod (3 < 6)	WL = 1. complete the weekly surveys 2. At Week 4 learn the same yoga practices
Li et al., 2022 [61]	1. Baduanjin	NR	NR Q I	12 weeks	45 min	≥5	Mod (3 < 6)	Learning health knowledge
Hamed et al., 2021 [62]	1. treadmill running 2. high-pace stationary cycling 3.weight-bearing aerobic exercises@HR (70% to 90%)	Aerobic	S NR I	8 weeks	80 min	5	Vig (6 < 9)	CBT
Chawla et al., 2022 [63]	Do static and dynamic squats from different angles + WBV (Whole-Body Vibrating Platform)	NR	U NR I	4 weeks	30 min	2	Vig (6 < 9)	Do static and dynamic squats from different angles
BarĞi, et al., 2021 [64]	1. 5000–10,000 steps/day 2. Doing leisure activities and/or doing housework	Aerobic	S Q I	4 weeks	30–60 min	3	Mod (3 < 6)	NT
Philippot, Arnaud et al., 2022 [65]	HIIT@HRmax (80% < 90%)	Aerobic	S Q I	4 weeks	≥10 min	3	Vig (6 < 9)	NT

Supervised (S) or unsupervised (U), qualied instructor (Q), group (G) or individual (I); NT, Not doing regular exercise training; WL, wait-list control; HR, max heart rate reduce age; * MET, metabolic equivalent estimate; NR, not reported or unclear.

**Table 4 ijerph-19-15338-t004:** Covariate Regression Analysis of Physical Activity on University Students’ Anxiety.

_ES	Coef.	Std. Err.	t	*p* > |t|	[95% Conf. Interval]
Met	−0.7531876	0.3061729	−2.46	0.091	−1.727567 to 0.2211914
Frequency (week)	−0.0706776	0.129785	−0.54	0.624	−0.4837114 to 0.3423562
Time (min)	−0.1395257	0.1319966	−1.06	0.368	−0.559598 to 0.2805465
Duration (weeks)	0.2457894	0.2962629	0.83	0.468	−0.6970513 to 1.18863
_cons	0.5127189	0.8040718	0.64	0.569	−2.046196 to 3.071634

**Table 5 ijerph-19-15338-t005:** Covariate Regression Analysis of Physical Activity on University Students’ Depression.

_ES	Coef.	Std. Err.	t	*p* > |t|	[95% Conf. Interval]
Met	−0.8071271	0.4329666	−1.86	0.203	−2.670032 to 1.055778
Frequency (week)	0.9779419	0.6021712	1.62	0.246	−1.612992 to 3.568875
Time (min)	0.0014643	0.2964396	0.00	0.997	−1.274012 to 1.276941
Duration (weeks)	0.0314791	0.4857233	0.06	0.954	−2.05842 to 2.121378
_cons	−0.6500254	1.14621	−0.57	0.628	−5.581768 to 4.281717

**Table 6 ijerph-19-15338-t006:** Subgroup analysis of physical activity on anxiety disorder of university students.

Group	Sub-Group	K	N	SMD	95% CI	*p*	I^2^
Met	3 < 6	5	919	−0.171	−0.301 to −0.040	0.010	0.0%
	6 < 9	3	276	−1.124	−1.387 to −0.860	0.000	0.0%
Frequency (week)	<4	6	754	−0.473	−0.907 to −0.038	0.033	85.1%
	≥4	2	441	−0.626	−1.558 to 0.306	0.188	89.3%
Time (min)	≤30	2	58	−0.598	−1.127 to −0.070	0.027	0.0%
	30 < 60	4	883	−0.401	−0.914 to 0.112	0.126	91.3%
	≥60	2	254	−0.673	−1.515 to 0.169	0.117	85.7%
Duration (weeks)	≤8	5	335	−−0.807	−1.213 to −0.402	0.017	57.8%
	>8	3	860	−0.163	−0.298 to −0.029	0.000	0.0%

K, number of trials; N, number of participants; SMD, standardized mean difference; CI, confidence interval.

**Table 7 ijerph-19-15338-t007:** Subgroup analysis of physical activity on college students’ depression.

Group	Sub-Group	K	N	SMD	95% CI	*p*	I^2^
Met	3 < 6	4	532	−0.419	−0.592 to −0.246	0.000	0.0%
	6 < 9	3	276	−0.859	−1.634 to −0.085	0.030	85.0%
Frequency (week)	<4	6	754	−0.674	−1.088 to −0.260	0.001	83.0%
	≥4	1	54	−0.273	−0.809 to 0.263	0.319	0.0%
Time (min)	≤30	2	58	−0.798	−1.335 to −0.261	0.004	0.0%
	30 < 60	3	496	−0.709	−1.467 to 0.048	0.067	91.8%
	≥60	2	254	−0.349	−0.597 to −0.101	0.006	0.0%
Duration (weeks)	≤8	5	335	−0.716	−1.286 to −0.145	0.014	79.1%
	>8	2	473	−0.417	−0.600 to −0.233	0.000	0.0%

K, number of trials; N, number of participants; SMD, standardized mean difference; CI, confidence interval.

## Data Availability

Not applicable.

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
