# Peer review of "Intervention of Physical Activity for University Students with Anxiety and Depression during the COVID-19 Pandemic Prevention and Control Period: A Systematic Review and Meta-Analysis"

_ijerph, 2022, doi:10.3390/ijerph192215338_

Round 1

Reviewer 1 Report

I would like to congratulate the team of authors for the complex study carried out on a relevant and, at the same time, necessary topic such as the assessment of the impact of physical activity on anxiety and depression in university students during the period of prevention and control of COVID-19, through a Systematic Review and Meta-analysis.

It represents a further step in the study and update of the effects produced by COVID-19 and the possible interventions for its reduction, as is the case of physical exercise.

After reviewing the article, I note my suggestions and recommendations:

-The abstract should conform to a maximum of 200 words.

-The structured research question is missing.

-Lines 130 to 132 are not applicable.

-"Reproducible studies" is striking to include them as an exclusion criterion.

-It is recommended to insert the titles and headings of the tables in all the pages they occupy.

Kind regards.

Author Response

Dear Reviewer 1,

Thank you for your kind comments on our article, which were very precise and helpful for us. In accordance with your suggestions, we have made a substantial revision of this article. Below, you will find a breakdown of the responses to your comments (in italics):

-The abstract should conform to a maximum of 200 words.

Response: Thank you for noting this.We have revised our abstract accordingly.

-The structured research question is missing.

Response: Thank you very much for your suggestion. We have added a structured research question..

-Lines 130 to 132 are not applicable.

Response: Thank you very much for your suggestion. We have deleted contents from lines 130 to 132.

-"Reproducible studies" is striking to include them as an exclusion criterion.

Response: Thank you very much for your suggestion.We have revised it to be a ‘repeated study’.

-It is recommended to insert the titles and headings of the tables in all the pages they occupy.

Response:Thank you very much for your suggestion. We have inserted the title and serial number of the tables in the pages where they are.

We tried our best to improve the manuscript and made some changes in the manuscript. These changes will not influence the content and framework of the paper. And here we did not list the changes but marked them in the revised version of the paper.

We hope you are pleased with the revised manuscript, which also includes the changes expected from other peer reviewers.

Thanks again, and kind regards.

Reviewer 2 Report

I read the article in depth, and in general, I think it is an interesting and important article. A significant part of the article is the international aspect it presents.

The main aspect missing from the article is a theory or theories that would organize the findings. This point can be addressed in two ways.

1. Choose a theory such as a theory of emotional regulation and its way of talking about the role of sports activities as a modulating variable of stress.

2. To present the theories used by the reviewed studies and create a separate category referring to the theoretical aspect of the articles.

3. The theoretical part is also important in discussing the findings.

4. Finally, I would like to see a detailed reference to theoretical and practical implications in the study conclusions.

Author Response

Dear Reviewer 2

Thank you for your kind comments on our article, which were very precise and helpful for us. In accordance with your suggestions, we have made a substantial revision of this article. Below, you will find a breakdown of the responses to your comments (in italics):

  1. Choose a theory such as a theory of emotional regulation and its way of talking about the role of sports activities as a modulating variable of stress.

Response: Thank you very much for your suggestion. We have added the emotion regulation theory in the main text and pointed out the role of physical activity as a mediator of emotion regulation.

  1. To present the theories used by the reviewed studies and create a separate category referring to the theoretical aspect of the articles.

Response: Thank you very much for your suggestion. The articles we borrowed in this study only refer to the experimental study of RCT, and the characteristics and measures are listed in the chart. According to suggestions given by you, we mainly adopted the emotion regulation theory of Opinion 1 in our revised version as a theoretical support of this article.

  1. The theoretical part is also important in discussing the findings.

Response: Thank you very much for your suggestion. in research findings, we have added some relevant content related to how emotional regulation works on mental health.

  1. Finally, I would like to see a detailed reference to theoretical and practical implications in the study conclusions.

Response: Thank you very much for your suggestion. We have added the relationship of physical activity and emotion regulation theory in the conclusion of the article, and listed the practical and theoretical influence of physical activity on emotion regulation theory.

We tried our best to improve the manuscript and made some changes in the manuscript. These changes will not influence the content and framework of the paper. And here we did not list the changes but marked them in the revised version of the paper.

We hope you are pleased with the revised manuscript, which also includes the changes expected from other peer reviewers.

Thanks again, and kind regards.